# The relationship between mental health and risk of active tuberculosis: a systematic review

Sally E Hayward ![ORCID] ,[1,2] Anna Deal ![ORCID] ,[1,2] Kieran Rustage ![ORCID] ,[1]
Laura B Nellums ![ORCID] ,[1,3] Annika C Sweetland,[4] Delia Boccia,[2] Sally Hargreaves ![ORCID] ,[1]
Jon S Friedland ![ORCID] [1]

DB, SH and JSF are joint senior authors.

¹Institute for Infection and Immunity, St George's, University of London, London, UK
²Faculty of Public Health and Policy, London School of Hygiene and Tropical Medicine, London, UK
³Division of Epidemiology and Public Health, School of Medicine, University of Nottingham, Nottingham, UK
⁴Department of Psychiatry, Columbia Vagelos College of Physicians and Surgeons, New York State Psychiatric Institute, New York, New York, USA

**Correspondence to**
Sally E Hayward;
shayward@sgul.ac.uk

## ABSTRACT

**Objectives** Tuberculosis (TB) and mental illnesses are highly prevalent globally and often coexist. While poor mental health is known to modulate immune function, whether mental disorders play a causal role in TB incidence is unknown. This systematic review examines the association between mental health and TB disease risk to inform clinical and public health measures.

**Design** Systematic review, following Preferred Reporting Items for Systematic Reviews and Meta-Analyses (PRISMA) guidelines.

**Search strategy and selection criteria** MEDLINE, PsycINFO and PsycEXTRA databases were searched alongside reference list and citation searching. Inclusion criteria were original research studies published 1 January 1970–11 May 2020 reporting data on the association between mental health and TB risk.

**Data extraction, appraisal and synthesis** Data were extracted on study design and setting, sample characteristics, measurement of mental illness and TB, and outcomes including effect size or prevalence. Studies were critically appraised using Critical Appraisal Skills Programme (CASP) and Appraisal Tool for Cross-Sectional Studies (AXIS) checklists.

**Results** 1546 records published over 50 years were screened, resulting in 10 studies included reporting data from 607 184 individuals. Studies span across Asia, South America and Africa, and include mood and psychotic disorders. There is robust evidence from cohort studies in Asia demonstrating that depression and schizophrenia can increase risk of active TB, with effect estimates ranging from HR=1.15 (95% CI 1.03 to 1.28) to 2.63 (95% CI 1.74 to 3.96) for depression and HR=1.52 (95% CI 1.29 to 1.79) to RR=3.04 for schizophrenia. These data align with evidence from cross-sectional studies, for example, a large survey across low-income and middle-income countries (n=242 952) reports OR=3.68 (95% CI 3.01 to 4.50) for a depressive episode in those with TB symptoms versus those without.

**Conclusions** Individuals with mental illnesses including depression and schizophrenia experience increased TB incidence and represent a high-risk population to target for screening and treatment. Integrated care for mental health and TB is needed, and interventions tackling mental illnesses and underlying drivers may help reduce TB incidence globally.

**PROSPERO registration number** CRD42019158071.

## Strengths and limitations of this study

► This review examines all available evidence on the relationship between mental health and tuberculosis (TB) incidence holistically.
► Comprehensive systematic review methods were used, following Preferred Reporting Items for Systematic Reviews and Meta-Analyses (PRISMA) guidelines.
► There was considerable variation in study design, which limits comparability of results across the included studies.

## INTRODUCTION

Both tuberculosis (TB) and mental health are urgent global health priorities, with 1.4 million TB deaths worldwide in 2019,[1] and approximately 14% of the global burden of disease attributable to neuropsychiatric disorders.[2] There is increasing recognition that physical and mental health are interconnected.[3 4] Mental illnesses are highly prevalent among TB patients and vice versa,[5 6] and poor mental health is associated with reduced treatment-seeking and adherence, and therefore with greater morbidity, mortality, transmission and drug resistance.[7 8] Moreover, TB can infect the central nervous system (CNS) causing neurological symptoms,[9] and certain anti-TB medications have psychiatric side effects.[5] The greatest burden of TB is experienced in individuals with risk factors including homelessness, drug and alcohol misuse, and migration.[10] Mental illness is more prevalent in all of these groups.[11–13] The relationships between TB and mental health are highly complex, with TB–depression comorbidity termed a 'syndemic' due to the bidirectional synergies involved.[14]

Previous research has examined the impact of TB and its treatment on mental health,[5] the relationship between mental disorders and TB treatment outcomes[7 15] and adherence,[8 16]

and the prevalence of mental disorders among TB and multidrug-resistant TB (MDR-TB) patients.[17–19] There is increasing interest in whether mental health may increase TB risk, supported by a growing body of evidence that chronic stressors and poor mental health directly influence the immune system, including susceptibility to infection.[14 20] How these pathways operate in TB is unknown, although various mechanisms are possible; for instance, the suppression of cellular immunity due to poor mental health could contribute to reactivation of latent TB infection (LTBI) or progression from subclinical to clinical disease.[21] However, while the bidirectional relationship between mental health and TB has been the subject of discussion,[6 14 21–25] there has been no systematic and comprehensive examination of the evidence base on associations between mental health and TB incidence. This systematic review aims to examine the evidence on the relationship between mental health and TB, to provide insight into whether poor mental health may be a risk factor for TB disease and to inform clinical and global public health measures.

## METHODS
### Search strategy
We carried out a systematic review following Preferred Reporting Items for Systematic Reviews and Meta-Analyses (PRISMA) guidelines, registered with PROS-PERO (CRD42019158071)[26]. MEDLINE, PsycINFO and PsycEXTRA were searched from inception to 11 May 2020, combining terms and subject headings for mental health and TB (online supplemental box 1), restricting the search to English language papers. Records were imported into EndNote, and duplicates were deleted. Two independent reviewers (SEH and KR/AD) carried out title/abstract and full-text screening using Rayyan QCRI.[27] We searched reference lists and carried out citation searching via Web of Science for included papers and previous reviews in this area.[5 7 8 15–19]

### Selection criteria
This study includes empirical research reporting on the relationship between poor mental health and risk of active TB (table 1). This includes two study designs: (1) longitudinal studies investigating the causal relationship between mental health and TB incidence and (2) cross-sectional studies investigating the association between mental health and TB disease. Cross-sectional studies were included regardless of whether they treated mental health as the exposure and TB as the outcome or vice versa, since both provide evidence for an association (provided that those with a history of mental illness are not excluded). Prevalence studies were excluded, as studies without controls cannot establish the presence of an association. No exclusions were made based on population, age or geographic location.

The exposure measure was mental ill health, assessed either as an overall measure of mental illness or as a specific mental disorder. Mental disorders are defined here to include psychotic disorders (eg, schizophrenia), mood or affective disorders (eg, depression), and neurotic, stress-related and somatoform disorders (eg, anxiety).[28] Mental illness may be diagnosed clinically (ie, by a medical professional and/or prescription of

**Table 1** Inclusion criteria, using PECOS framework

| | Inclusion criteria | Exclusion criteria |
|---|---|---|
| Population | Any population in any geographic location, including vulnerable groups, for example, the homeless | |
| Exposure | Mental illness, including psychotic and affective disorders, diagnosed clinically or by any psychological tool | ► Alcohol and drug disorders.<br>► Studies where mental illness is not a primary variable. |
| Control | No mental illness | |
| Outcome | Incidence of active TB in humans, including reactivation of LTBI and MDR-TB | ► LTBI without reactivation.<br>► CNS TB (including tuberculosis meningitis).<br>► TB-HIV coinfection.<br>► Studies investigating outcomes, for example, mortality (and not incidence). |
| Study design | Observational epidemiological studies (a) longitudinal studies (cohort): mental health must be treated as the exposure and TB as the outcome; (b) cross-sectional studies (cross-sectional and case–control): mental health may be treated as the exposure and TB as the outcome or vice versa | ► Prevalence studies without controls.<br>► Qualitative studies.<br>► Case reports.<br>► Non-original research, for example, protocols, commentaries, reviews.<br>► Longitudinal studies in which TB is the exposure and mental health is the outcome.<br>► Cross-sectional studies that exclude those with a history of mental illness. |
| Dates | Papers published 1970–2020 | |

CNS, central nervous system; LTBI, latent tuberculosis infection; MDR-TB, multi-drug resistant tuberculosis; PECOS, population, exposure, control, outcome, study design; TB, tuberculosis.

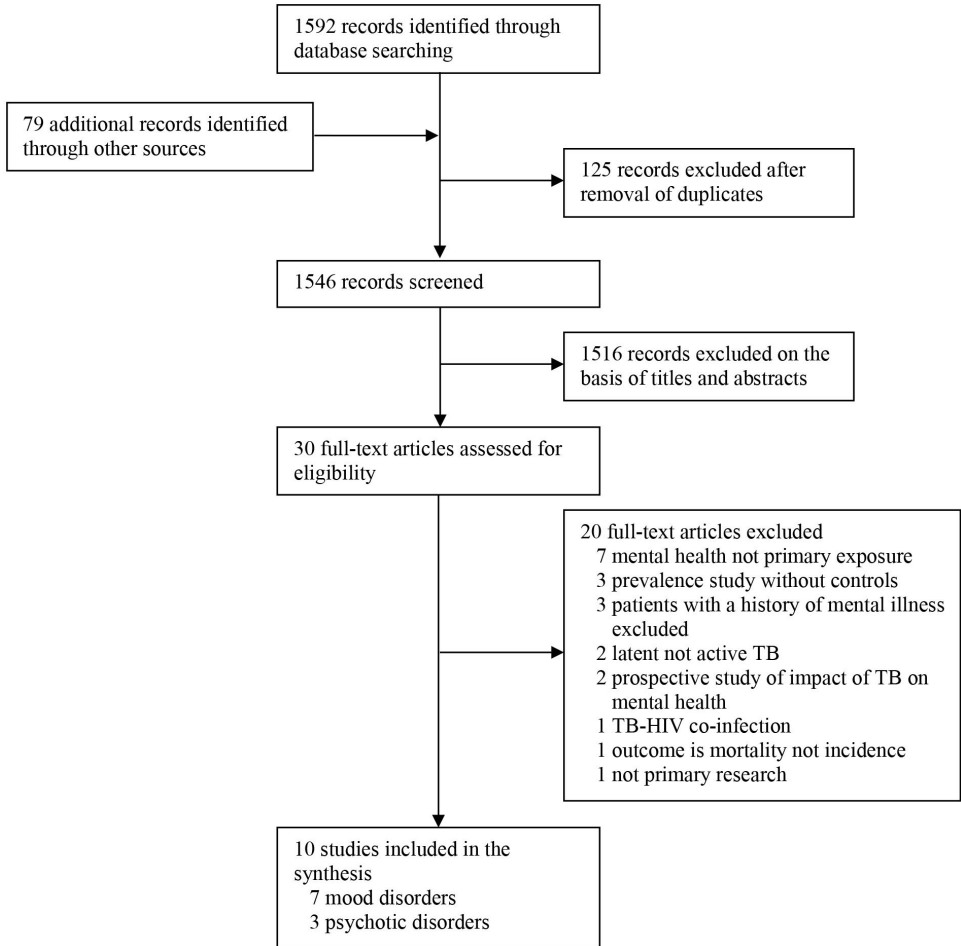

**Figure 1** Study selection, PRISMA flow chart. PRISMA, Preferred Reporting Items for Systematic Reviews and Meta-Analyses; TB, tuberculosis.

medication) or by any psychological tool (ie, a structured questionnaire or interview). Only studies for which mental illness was a primary variable were included to ensure that the study was powered to detect an association. Studies focused solely on substance use disorders were excluded as alcohol and drug use are already known to be strongly associated with TB risk, possibly due to social mixing patterns, or the effect of these substances on the immune system.[29–31]

The outcome measure was active TB disease, diagnosed according to bacteriological, clinical and/or radiological criteria.[32] This excludes LTBI but includes reactivated disease. TB of any site, pulmonary or extrapulmonary, was included, with the exception of CNS TB. TB-HIV coinfection was also excluded. In these cases, different mechanisms are expected to underlie the associations. Studies relating to outcomes of TB infection or treatment (including psychiatric side effects of anti-TB medication), rather than incidence, were excluded, as this has been previously reviewed.[7 15]

### Data extraction

Two independent reviewers (SEH and KR/AD) extracted data using a standardised form adapted from the Cochrane Effective Practice and Organisation of Care

(EPOC) group,[33] including study characteristics, population, setting, methods, participants, exposure, outcome, results (summary estimates), and applicability.

### Critical appraisal

Two independent reviewers (SEH and KR/AD) assessed the quality of all included studies, using Critical Appraisal Skills Programme (CASP) checklists for cohort and case–control studies and the Appraisal Tool for Cross-Sectional Studies (AXIS) tool for cross-sectional studies.[34 35] The tools were amended to remove questions on local applicability and implications for practice, instead asking about external validity and implications for the review. A study was defined as high quality at a critical appraisal score of above 90%, moderate above 60% and low at 60% or below. This score relates to the appropriateness and relevance of the study for this review question rather than in relation to the study's own aims.

### Synthesis

Extracted data were tabulated, detailing methods and results of included studies, categorised by type of mental illness. Results are presented as reported in the studies. The primary outcome extracted is the main effect measure for the relationship between mental health and TB,

**Table 2** Characteristics of all included studies

| | Population | | Study design | | | | Exposure assessment | | Outcome assessment | | Controls | Critical appraisal* |
|---|---|---|---|---|---|---|---|---|---|---|---|---|
| | Country | Gender and age | Design | Setting | Study period | N | Mental disorder | Measurement | Site of TB | Measurement | | |
| **Mood disorders** | | | | | | | | | | | | |
| *Longitudinal studies* | | | | | | | | | | | | |
| Oh et al[36] | South Korea | 34% male; mean 45.8 years (SD 18.4) | Retrospective cohort study | Nationwide database | 2003–2013 | 64 744 | Depression | ICD-10 codes plus psychotherapy prescription | Any | ICD-10 codes plus anti-TB drug prescription | No mood disorders, matched by age and sex | High (10/11, 91%) |
| Cheng et al[37] | Taiwan | 62% male; mean 47.9 years (SD 16.5) in cases and 47.6 (16.6) in controls | Retrospective cohort study | Nationwide database | 2000–2013 | 172 952 | Depression | ICD-9 codes | Pulmonary | ICD-9 codes | No depression, matched for age, sex and comorbidities | High (10/11, 91%) |
| *Cross-sectional studies* | | | | | | | | | | | | |
| Koyanagi et al[45] | LMICs | 49% male; mean 38.4 years (SD 16.1) | Cross-sectional study (population-based) | Community-based in 48 countries | 2002–2004 | 242 952 | Depression | Interview (World Mental Health Survey version of the CIDI) | Pulmonary | TB symptoms in past 12 months (cough for 3 weeks or longer, blood in phlegm) | N/A | Moderate (17/20, 85%) |
| Castro-Silva et al[41] | Brazil | 63% male in cases, 61% in controls; mean 40.7 years (SD 15.7) in cases, 46.9 (SD 16.0) in controls | Cross-sectional study (comparing point prevalence in cases and controls) | Municipal health centre in Rio de Janeiro | 2015–2016 | 260 | Depression | Questionnaire (PHQ-9) and interview (MINI-Plus) | Pulmonary | Smear microscopy and/or Xpert MTB/RIF | Patients without TB, who were suspected to have pulmonary TB before testing | Moderate (13/20, 65%) |
| de Araújo et al[42] | Brazil | 61% male; mean 38.2 years (SD 14.2) | Case–control study | 3 referral hospitals and 6 community clinics in Salvador | 2008–2010 | 1434 | Common Mental Disorders | Questionnaire (SRQ-20) | Pulmonary | Smear microscopy and culture for *M.tb* | Non-TB symptomatic respiratory patients, age-matched and sex-matched | Moderate (8/10, 80%) |
| Hernández Sarmiento et al[43] | Columbia | 83% male; mean 38.9 years (SD 10.4) | Cross-sectional study (population-based) | Homeless population at a local health facility in Medellin | 2006–2007 | 426 | Mental disorders | Interview (MINI) | Pulmonary | Smear microscopy | N/A | Low (12/20, 60%) |
| Srivastava et al[38] | India | 67% male; age not available | Cross-sectional study (comparing point prevalence in cases and controls) | Hospital for TB and chest diseases, Bikaner | Not stated | 120 | Psychological state | Interview (PSE) | Pulmonary | Not specified | Patients with non-TB at the hospital | Low (5/20, 25%) |
| **Psychotic disorders** | | | | | | | | | | | | |
| *Longitudinal studies* | | | | | | | | | | | | |
| Kuo et al[39] | Taiwan | 55% male; median 35.4 years in cases, 35.3 in controls | Retrospective cohort study | Nationwide database | 1998–2009 | 120 818 | Schizophrenia | ICD-9 codes | Any | ICD-9 codes plus anti-TB drug prescription | No schizophrenia, matched for age, sex, index date and comorbidities | High (10/11, 91%) |

Continued

**Table 2** Continued

| Country | Gender and age | Design | Study period | Setting | N | Mental disorder | Measurement | Site of TB | Measurement | Controls | Critical appraisal* |
|---|---|---|---|---|---|---|---|---|---|---|---|
| | | | | | | | | | | | |
| Ohta et al[40] | Japan | 55% male; age not available | Retrospective cohort study | 1960–1978 | Nationwide database | 3251 | Schizophrenia | Diagnosis (not further specified) | Any | Diagnosis (not further specified) | Expected incidence based on annual incidence rate | Moderate (7/11, 64%) |
| *Cross-sectional study* | | | | | | | | | | | |
| Lasebikan and Ige[44] | Nigeria | 38% male in cases, 18% in controls; median 35 years in cases, 42 in controls | Cross-sectional study (comparing point prevalence in cases and controls) | 2010–2014 | MDR-TB treatment centre, Ibadan | 227 | Psychosis | Questionnaire (GHQ-12) and interview (psychosis screening questionnaire plus Structured Clinical Interview for DSM-IV Axis I Disorder, psychosis module) | Pulmonary | Attendance at MDR-TB treatment centre (not further specified) | Accompanying family members or caregivers | Low (11/20, 55%) |

*Critical appraisal score is defined based on number of positive responses to the questions on Critical Appraisal Skills Programme (CASP) checklists for cohort and case–control studies and the Appraisal Tool for Cross-Sectional Studies (AXIS) for cross-sectional studies.

CIDI, Composite International Diagnostic Interview; CMD, common mental disorder; DSM, diagnostic and statistical manual; GHQ, General Health Questionnaire; ICD, International Classification of Diseases; LMIC, low-income and middle-income country; MDR, multi-drug resistant; MINI, Mini-International Neuropsychiatric Interview; *M.tb/MTB, Mycobacterium tuberculosis*; N, sample size; PHQ, Patient Health Questionnaire; PSE, present state interview; RIF, rifampicin; SRQ, Self-Reporting Questionnaire; TB, tuberculosis.

and the secondary outcome is any additional measures reported relating to different definitions or levels of mental illness. For cross-sectional studies, any cases of mental illness arising post-TB diagnosis or treatment were excluded where possible. Given the wide range of populations, study designs and statistical methods used, meta-analysis was not appropriate. A narrative synthesis was therefore completed, by study design and type of mental illness. Effect size and precision were compared where studies were sufficiently similar. The evidence as a whole was assessed as to whether poor mental health may be associated with, and act as a risk factor for, TB disease.

### Patient and public involvement

Patients and/or the public were not involved in the design, conduct, reporting or dissemination plans of this research.

### RESULTS

We screened 1546 records published over 50 years and found that 10 published articles met the inclusion criteria (figure 1), with a combined sample size of 607 184. Reasons for exclusion are presented in figure 1, with the most common being that mental health is not a primary outcome or that the study design does not meet the inclusion criteria outlined above.

Included studies span multiple locations, including five in Asia,[36–40] three in South America,[41–43] one in Africa[44] and one across low-income and middle-income countries (LMICs).[45] Two studies were published in the 1980s[38 40] and the remainder within the last 10 years.[36 37 39 41–45] Seven studies assess mood disorders[36–38 41–43 45] and three assess psychotic disorders.[39 40 44] Seven studies investigate pulmonary TB exclusively,[37 38 41–45] and three consider both pulmonary and extrapulmonary disease.[36 39 40] Study characteristics are summarised in table 2. Only one paper included here was included in any prior systematic review of mental health and TB.[5 7 8 15–19]

Several study designs were used to investigate the association between mental health and TB. Four retrospective cohort studies use insurance or registry data to follow-up TB incidence in those with a mental illness compared with matched controls[36 37 39] or general population incidence.[40] Where specified, the studies use International Classification of Diseases (ICD-9 or ICD-10) codes on patient medical records, in addition to the prescription of medication, to identify patients with a mental disorder and with TB disease. These are large studies, making up 59.6% of the total sample size (n=361 765) (figure 2).

Other studies use questionnaire screening tools or interviews to assess mental health at a single timepoint. These may be short disorder-specific questionnaires, such as the Patient Health Questionnaire-9 (PHQ-9) for depression,[46] or longer interviews covering a range of psychiatric disorders, such as the Composite International Diagnostic Interview (CIDI).[47] One case–control study identifies mental illnesses prior to TB diagnosis in cases and

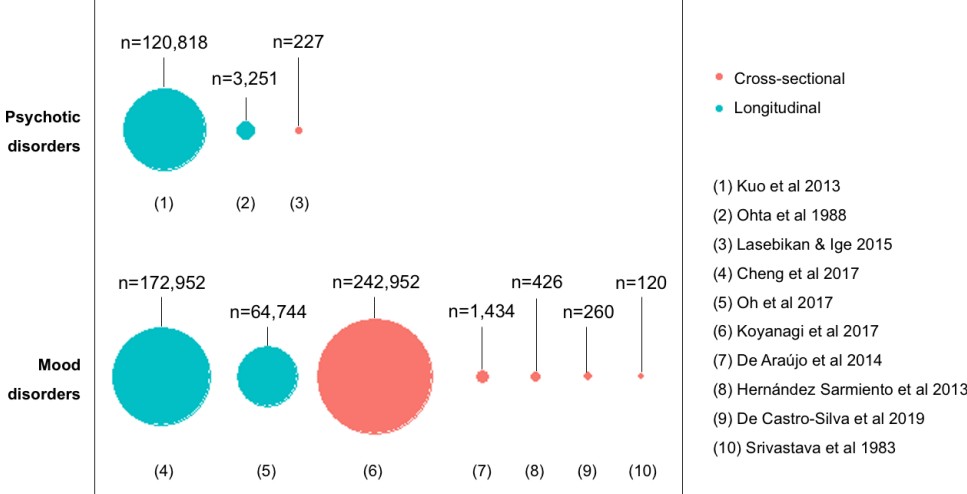

**Figure 2** Sample size of included studies.

matched controls.[42] Two population-based cross-sectional studies establish prevalence of mental illnesses and TB, and assess the association between them.[43 45] Three additional cross-sectional studies compare point prevalence of mental illnesses in TB cases and controls.[38 41 44] One of the population-based cross-sectional studies is very large, making up 40.0% of the total sample size (n=242 952), whereas the other five cross-sectional studies are smaller, making up 0.4% of the total sample size (n=2467) (figure 2).

While the retrospective cohort studies use population-based databases or registers,[36 37 39 40] other studies are only applicable to certain populations such as the homeless patients[43] or patients with MDR-TB.[44] Included studies vary in quality, with three high, four moderate and three low-quality studies overall, and an average critical appraisal score of 71% for mood and 70% for psychotic disorder papers. The longitudinal studies are all moderate to high quality (average score 84%), whereas the cross-sectional studies are low to moderate quality (average score 62%). Common problems identified are issues with sampling and inadequate consideration of confounding. Results and study quality are tabulated in table 3 and in detail in online supplemental tables 1,2.

### Mood disorders

Four studies specifically investigate depression,[36 37 41 45] and three look generally at mood disorders.[38 42 43] This includes two large, high-quality cohort studies on depression, both using nationally representative databases. One, in South Korea, reports an adjusted HR for TB incidence among those with depression compared with controls of 2.63 (95% CI 1.74 to 3.96, p<0.001),[36] and the other, in Taiwan, reports an adjusted HR of 1.15 (95% CI 1.03 to 1.28).[37] The former also finds a dose–response relationship, with more severe depression associated with greater TB risk. Both studies report a higher TB incidence rate in those with depression than in controls (figure 3).

Other studies use a cross-sectional design to assess the association between mood disorders and TB. A large study using community-based World Health Survey data from across LMICs reports an adjusted OR for a depressive episode in those with TB symptoms vs those without of 3.68 (95% CI 3.01 to 4.50).[45] By contrast, a cross-sectional study among presumptive pulmonary TB cases in Rio de Janeiro, Brazil that compares the prevalence of depression among patients with and without a bacteriologically confirmed diagnosis of pulmonary TB finds no evidence of a difference. Of 259 participants screened for depression, the prevalence among TB cases is 60.2% versus 62.1% in controls (crude OR=0.92, 95% CI 0.55 to 1.54, p=0.79), and of 159 who screened positive for depression, this diagnosis was confirmed in 59.5% of TB cases versus 50.9% of controls (crude OR=1.42, 95% CI 0.63 to 3.19, p=0.42).[41]

Beyond depression, a case–control study in Salvador, Brazil considers common mental disorders (CMDs) more broadly, characterised by diverse depressive, anxiety or somatoform symptoms, and finds elevated odds of having TB among those with a CMD (adjusted OR=1.34, 95% CI 1.05 to 1.70).[42] A cross-sectional study among the homeless in Medellín, Columbia investigates the association between TB and various psychiatric illnesses, but finds that only dysthymia (persistent mild depression) and history of major depression are associated with TB, and only for dysthymia is this statistically significant in multivariate analysis (adjusted OR=2.54, 95% CI 1.10 to 5.86, p=0.028).[43] Finally, a cross-sectional study into the current psychological state of TB cases and controls in Bikaner, India finds a significant association between TB and symptoms including anxiety, depressed mood and sleep/appetite disturbances, and reports a prevalence of psychiatric illness in TB patients of 41.6% versus 13.3% in controls.[38]

### Psychotic disorders

Two studies specifically investigate schizophrenia,[39 40] and one considers psychosis more generally.[44]

**Table 3** Summary of study findings

| | Primary outcome | | Secondary outcomes | | Statistical method | Adjustment |
|---|---|---|---|---|---|---|
| | Effect measure | Results | Effect measure | Results | | |
| **Mood disorders** | | | | | | |
| *Longitudinal studies* | | | | | | |
| Oh et al[36] | Adjusted HR for TB incidence in depressed vs controls | 2.63 (95% CI 1.74 to 3.96, p<0.001) | Adjusted HRs stratified by mild and severe depression | For mild depression=1.99 (95% CI 1.21 to 3.28, p=0.007), for severe depression=3.08 (95% CI 2.00 to 4.73 p<0.0001); linear dose-response | Cox proportional hazards model | Age, sex, income level, DM, COPD, alcoholism |
| Cheng et al[37] | Adjusted HR for pulmonary TB incidence in depressed vs controls | 1.15 (95% CI 1.03 to 1.28) | – | – | Cox proportional hazards model | Age, sex, alcohol-related disease, CKD, chronic liver disease, COPD, DM, HIV infection, gastrectomy and pneumoconiosis |
| *Cross-sectional studies* | | | | | | |
| Koyanagi et al[45] | Adjusted OR for depressive episode in those with pulmonary TB vs those without | 3.68 (95% CI 3.01 to 4.50, p<0.0001) | Adjusted ORs for brief depressive episode and for subsyndromal depression | For brief depressive episode=1.75 (95% CI 1.26 to 2.42, p=0.0008, for subsyndromal depression=1.98 (95% CI 1.47 to 2.67, p<0.0001) | Logistic regression | Age, sex, education, wealth, household size, setting, current smoking, alcohol consumption, BMI, diabetes, country |
| Castro-Silva et al[41] | Prevalence of depression in pulmonary TB patients; crude OR | Screening tool: 60.2% vs 62.1% in controls (OR=0.92, 95% CI 0.55 to 1.54, p=0.79) Diagnostic tool: 59.5% vs 50.9% in controls (OR=1.42, 95% CI 0.63 to 3.19, p=0.42) | – | – | Logistic regression | – |
| de Araújo et al[42] | Adjusted OR for pulmonary TB in those with a CMD vs those without | 1.34 (95% CI 1.05 to 1.70) | – | – | Logistic regression | Diabetes, alcohol abuse, ethnicity, drug use, number of household goods, level of education, history of contact and crowding |
| Hernández Sarmiento et al[43] | Crude and adjusted ORs for pulmonary TB in those with mental disorders | Dysthymia (adjusted)=2.54 (95% CI 1.10 to 5.86, p=0.028), dysthymia (crude)=2.66 (95% CI 1.19 to 5.19, p=0.013), previous history of major depression (crude)=2.22 (95% CI 1.07 to 4.6, p=0.028) | – | – | Logistic regression | Age, gender, time living on the streets, education level, marital status, interaction with other homeless people, income sources |
| Srivastava et al[38] | Prevalence of psychiatric illness in pulmonary TB patients vs controls | 41.6% vs 13.3% in controls (p<0.001) | Prevalence of at least one psychiatric symptom in pulmonary TB patients | 86.6% vs 70.0% in controls | $\chi^2$ test | – |
| **Psychotic disorders** | | | | | | |
| *Longitudinal studies* | | | | | | |
| Kuo et al[39] | Adjusted HR for TB incidence in those with schizophrenia vs controls | 1.52 (95% CI 1.29 to 1.79, p<0.001) | – | – | Cox proportional hazards model | Age, sex, Charlson's score, COPD, DM, rheumatoid disease, peptic ulcer disease, liver disease, hypertension, arrhythmia, dyslipidaemia, drug or substance abuse, CKD, cancer, heart failure, peripheral vascular disease, myocardial infarction, hemiplegia/paraplegia, AIDS |

Continued

**Table 3** Continued

| | Primary outcome | | Secondary outcomes | | Statistical method | Adjustment |
|---|---|---|---|---|---|---|
| | Effect measure | Results | Effect measure | Results | | |
| Ohta et al [40] | RR for TB incidence observed in those with schizophrenia vs expected based on general population rates | 3.04 (p<0.005) | – | – | Observed vs expected relative risk | Age, sex |
| **Cross-sectional study** | | | | | | |
| Lasebikan and Ige [44] | Prevalence of psychosis and schizophrenia in patients with MDR-TB vs controls | Psychosis: 33.0% (18.3% excluding medication-induced psychotic disorders*) vs 2.7% in controls. Schizophrenia: 8.7% vs 0.9% in controls ($\chi^2$=5.9, p=0.02) | Prevalence of psychiatric morbidity in patients with MDR-TB vs controls | 81.7% vs 51.7% in controls ($\chi^2$=21.7, p<0.001) | $\chi^2$ test | – |

*Recalculated to exclude mental health conditions that arise after TB diagnosis or treatment.
BMI, body mass index; CKD, chronic kidney disease; CMD, common mental disorder; COPD, chronic obstructive pulmonary disease; DM, diabetes mellitus; MDR, multi-drug resistant; RR, relative risk; SEP, socioeconomic position; TB, tuberculosis.

The schizophrenia studies are retrospective cohort studies, both finding that schizophrenia is significantly associated with TB risk. A large, high-quality study in Taiwan, using a nationwide database, reports an adjusted HR for TB incidence in those with schizophrenia versus controls of 1.52 (95% CI 1.29 to 1.79, p<0.001).[39] The other, in Nagasaki, Japan uses registry data to identify TB incidence in those diagnosed with schizophrenia, and compares this to general population incidence. The reported relative risk (RR) for observed versus expected TB incidence is 3.04 (p<0.005).[40] Both studies report a higher TB incidence rate in those with schizophrenia than in controls (figure 3).

A cross-sectional study investigates psychosis more generally among patients attending an MDR-TB clinic in Ibadan, Nigeria. This reports a prevalence of psychosis of 33.0% in patients with MDR-TB versus 2.7% in controls. When recalculated to exclude anti-TB medication-induced psychotic disorders, this is a prevalence of 18.3% in patients with MDR-TB. This includes a prevalence of schizophrenia of 8.7% in TB patients versus 0.9% in controls ($\chi^2$=5.9, p=0.02).[44]

## DISCUSSION

Included studies, synthesising data from 607 184 individuals, show an association between mental health and TB. This includes robust evidence from cohort studies, all of which were based in Asia, demonstrating that depression and schizophrenia increase risk of TB disease. While the coexistence of poor mental health and TB is consistent with existing literature,[5] this is the first review to show that mental health is a risk factor for active TB.

Four population-based cohort studies are included, which are large studies using registry or insurance records to assess the relationship between medically coded depression or schizophrenia and TB disease in nationwide samples in Asia. Other studies, which are generally much smaller and cross-sectional in design, use questionnaire or interview mental health screening tools carried out in community or healthcare settings across Asia, South America and Africa. They therefore can detect less severe and undiagnosed conditions and can focus on specific, vulnerable populations such as the homeless. Whereas the cohort studies consider specific conditions such as depression or schizophrenia, most of the cross-sectional studies use screening tools that capture a wider range of mental health conditions.

Despite significant heterogeneity in study design and population, nine out of 10 included studies find a significant association between mental health and TB. In the one study that does not find a significant association, controls are individuals with prolonged respiratory symptoms, who might be expected to have a higher prevalence of depression than the general population.[41] While not directly comparable, some of the observed variations may be attributed to different levels of adjustment for confounding; the cohort studies with more extensive

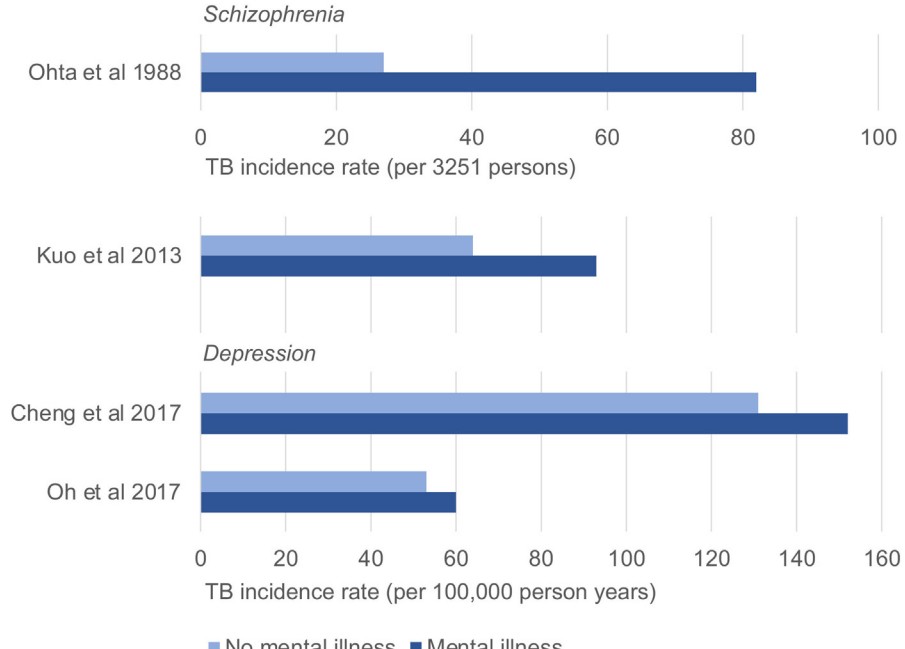

**Figure 3**  TB incidence rate reported in longitudinal studies. TB, tuberculosis.

adjustment for confounders[37 39] show smaller effect sizes than their counterparts with less or no adjustment for confounders.[36 40]

A key strength of this study is that, through undertaking an exhaustive search of the literature, it brings together a large sample of over 600 000 individuals. By synthesising evidence from a range of study designs, we collate all the evidence on this topic and analyse the findings holistically. Nevertheless, this review has certain limitations. Future reviews could focus on LTBI, CNS TB and TB-HIV coinfection, which were excluded from this study. Moreover, the heterogeneity in study populations and methods made drawing direct comparisons between studies difficult and made meta-analysis inappropriate. For example, it was not possible to directly compare measures of effect generated from rates (HRs) with those generated from risks (RRs/ORs).

The nature of available evidence means that some caution is required when drawing conclusions. Studies addressing the association between mental health and TB are of variable design and quality, with four large studies (three cohort and one cross-sectional) accounting for 99% of included participants.[36 37 39 45] Three included studies were classified as low quality; however, these only account for 0.001% of the total sample size (773 participants).[38 43 44] Only the cohort studies can establish temporality to provide compelling evidence that mental health precedes and acts as a risk factor for TB, and these are the highest quality studies.[36 37 39 40] However, the cohort studies only cover depression and schizophrenia and are all from Asia. Further cohort studies are required to confirm whether these findings hold true for other mental illnesses in other global regions.

The study findings increase our understanding of TB risk factors, yet further research is needed to elucidate the pathways by which mental health may increase TB incidence, causally or via associations between mental health and other risk factors for TB such as alcohol/drug use, homelessness, incarceration, physical comorbidities and poverty. The relationships between mental health, TB and social risk factors are multi-directional, meaning that complex conceptual frameworks will be needed to understand the observed associations.

The existence of plausible immune mechanisms supports a causal explanation, with evidence that psychosocial stressors are associated with immune biomarkers relevant to TB.[48] Mental illnesses including depression are associated with various immunological changes which could increase susceptibility to TB[14 49]; however, the key neuroendocrine and immunological pathways involved are unknown. In addition, mental health could be one factor influencing risk of progression from LTBI to active disease. Understanding the pathways that connect mental health and the immune response to TB may guide the development of host-directed approaches, for example, to prevent reactivation of LTBI.

The findings have implications for policy and clinical practice. The *Lancet* Commission on TB recommends that to achieve a TB-free world, populations at high risk must be reached and brought into care.[50] The evidence presented here suggests that those suffering from mental illnesses, in particular depression and schizophrenia, could constitute such a high-risk group for active case-finding and treatment. This group could benefit from a holistic approach, integrating services for mental health and TB to facilitate rapid diagnosis and treatment of TB disease and LTBI, as well as providing better mental health support for individuals with TB. As such, the WHO End TB Strategy 2015–2035 recommends that treatment for TB and mental health is brought together.[51] Providing high-quality mental health support and ensuring treatment adherence for TB both require substantial engagement with patients, so

leveraging this contact to provide holistic care could prove effective.[52]

In addition, the finding that mental illnesses constitute risk factors for TB suggests that tackling poor mental health and its underlying drivers may reduce TB incidence. In LMICs with a high TB incidence, poverty is consistently associated with common mental illnesses.[53] Social protection schemes that lift individuals out of poverty are known to improve mental health and reduce TB risk factors,[54–56] suggesting that tackling poverty and associated poor mental health through investment in wider social policies could help reduce TB incidence.[57]

## Conclusion

We find evidence that mental health is a risk factor for active TB. There is robust evidence from cohort studies based in Asia that depression and schizophrenia increase incidence of TB disease. This data, in combination with evidence from cross-sectional studies, identifies individuals with mental illnesses as a high-risk population for clinical TB that could be targeted for screening and treatment. This highlights the need for integrated programmes providing care for mental health and TB and suggests that interventions that tackle mental illnesses and their underlying drivers may help reduce TB incidence globally.

**Contributors**  SEH and DB developed the concept and study design. SEH, AD and KR carried out screening, data extraction, critical appraisal and analysis. SEH is responsible for the overall content as guarantor. All authors (SEH, AD, KR, LN, ACS, DB, SH, JSF) contributed to critical review of data and manuscript writing.

**Funding**  SEH and AD are supported by Medical Research Council PhD studentships (MR/N013638/1), and KR receives funding from the Rosetrees Trust (M775). SH is supported by the National Institute for Health Research (NIHR Advanced Fellowship NIHR300072) and the Academy of Medical Sciences (SBF005\1111), and the European Society for Clinical Microbiology and Infectious Diseases (ESCMID) through an ESCMID Study Group for Infections in Travellers and Migrants (ESGITM)/ESCMID Study Group for Mycobacterial Infection (ESGMYC) research grant. LN receives funding from the Academy of Medical Sciences (SBF005\1047), the Medical Research Council/Economic and Social Research Council/Arts and Humanities Research Council (MR/T046732/1), and the Medical Research Council (MR/V027549/1). ACS is supported by the US National Institute of Mental Health (K01 MH104514).

**Competing interests**  None declared.

**Patient consent for publication**  Not applicable.

**Ethics approval**  This study does not involve human participants.

**Provenance and peer review**  Not commissioned; externally peer reviewed.

**Data availability statement**  All data relevant to the study are included in the article or uploaded as supplementary information.

**ORCID iDs**
Sally E Hayward http://orcid.org/0000-0002-4105-0990
Anna Deal http://orcid.org/0000-0001-6168-6542
Kieran Rustage http://orcid.org/0000-0003-1599-7635
Laura B Nellums http://orcid.org/0000-0002-2534-6951
Sally Hargreaves http://orcid.org/0000-0003-2974-4348
Jon S Friedland http://orcid.org/0000-0001-7789-9649

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
