## [Reviewer comments · BMJ Open]

ARTICLE DETAILS

TITLE (PROVISIONAL)	The relationship between mental health and risk of active tuberculosis: A systematic review
AUTHORS	Hayward, Sally; Deal, Anna; Rustage, Kieran; Neilums, Laura; Sweetland, Annika; Boccia, Delia; Hargreaves, Sally; Friedland, Jon

VERSION 1 – REVIEW

REVIEWER	Azeez, A Department of Statistics, University of Fort Hare
REVIEW RETURNED	03-Apr-2021

GENERAL COMMENTS	1. The study included a peer review protocol and provides summaries of the available literature2. Clear objectives are identified3. Comprehensive search conducted from different databases4. There is a comprehensive evaluation of study in the process of evaluating articles quality from the results and data synthesis.
--

REVIEWER	Brooks, Meredith Harvard University, Global Health and Social Medicine
REVIEW RETURNED	19-Apr-2021

GENERAL COMMENTS	Thank you for the opportunity to review your article, “The relationship between mental health and risk of active tuberculosis: A systematic review.” This is an important, often overlooked, topic. I hope that the results of this review can help to inform more integrated screening and care that is desperately needed globally. Please see comments below: ABSTRACT: • The results state that 1,546 articles were screened and resulted in data from 607,184 individuals. It would be good to include the intermediary of how many articles were included that resulted in this number of individuals. Your unit of analysis is articles, not individuals, so focusing less on the number of individuals and more on the number of included articles is more important.• The authors mention that the study spans several continents but only report data from one; is there a reason for this?• Instead of just noting that there is evidence of an association from cross-sectional studies, can you present the evidence?• It would be good to also report confidence intervals. MAIN TEXT: • Please update reference #1 to 2020 Global TB Report.
---

	 • The methods section is severely lacking in regards to details about the main exposure and outcome measures. It is unclear whether mental health was assessed in aggregate or only by types of mental health disorders. The definition of these disorders and their mechanism for confirmation of diagnosis is also lacking. While Table 1 notes that the exposure should be diagnosed clinically or by any psychological tool, this is not in the methods anywhere and should be defined in more detail. Also, how the included articles actually diagnose and define the exposure and outcome should be included in the results section. Some of this information is included in Table 2 but it is important for the reader to have this information readily available to demonstrate similarities and differences across studies. This goes for both the exposure definitions and outcome measure definition. For example, it is unclear of whether you were only interested in studies with bacteriologically confirmed cases or not. Also, can studies include individuals of any age or only adults? If so, what age range is being used to define adults. • Table 1 says inclusion criteria are prospective cohort studies, but Table 2 shows that retrospective cohort studies were also included. • The results section notes that studies were assessed over a 50 year period, but how recent were the 10 included? Can the authors provide some more information in the text about reasons why articles were excluded? When the authors note that subgroups of included studies make up a combined sample size (lines 112, 120, 121), it would be useful to include the percentage of the total sample size that they make up (ex. (4/10) 40% of the studies included make up 60% (361,765) of the patients included). • Line 170—is there a 95% CI that can be reported? • Quality of studies was not adequately addressed in the text of the results section. • It is unclear why line 184 concentrates on studies in Asia when there were also studies from other continents included? • Table 2 shows the critical appraisal score from the CASP and AXIS tools; there is one score that is out of 10 while all others are out of 11 and 20. Should this be out of 11? • The discussion seems to have quite a bit of repeat reporting of results which seems unnecessary. It would be good to have more discussion of the observed similarities and differences across studies, including settings/population, measuring of the exposure and outcome variables, etc. and their implication on the observed results.
--	--

VERSION 1 – AUTHOR RESPONSE

Reviewer 1 (Dr. A Azeez, Department of Statistics, University of Fort Hare)

1. The study included a peer review protocol and provides summaries of the available literature
2. Clear objectives are identified
3. Comprehensive search conducted from different databases
4. There is a comprehensive evaluation of study in the process of evaluating articles quality from the results and data synthesis.

Thank you for your positive feedback.

Reviewer 2 (Dr. Meredith Brooks, Harvard University)

Thank you for the opportunity to review your article, "The relationship between mental health and risk of active tuberculosis: A systematic review." This is an important, often overlooked, topic. I hope that the results of this review can help to inform more integrated screening and care that is desperately needed globally. Please see comments below:

Thank you for your useful feedback, we agree that this is a topic with important implications for policy and practice.

ABSTRACT:

- The results state that 1,546 articles were screened and resulted in data from 607,184 individuals. It would be good to include the intermediary of how many articles were included that resulted in this number of individuals. Your unit of analysis is articles, not individuals, so focusing less on the number of individuals and more on the number of included articles is more important.

We have added that 10 studies were included (Abstract, Results, page 2).

- The authors mention that the study spans several continents but only report data from one; is there a reason for this?

We chose to focus on data from the cohort studies in the abstract, which were all from Asia. However, we have now added some data from a cross-sectional study spanning LMICs, see below (Abstract, Results, page 2).

- Instead of just noting that there is evidence of an association from cross-sectional studies, can you present the evidence?

We have now added data from the largest cross-sectional study, which takes place across LMICs, as an exemplar (Abstract, Results, page 2).

- It would be good to also report confidence intervals.

Confidence intervals have now been added where available (Abstract, Results, page 2).

MAIN TEXT:

- Please update reference #1 to 2020 Global TB Report.

This has now been updated (Introduction, lines 3-4).

- The methods section is severely lacking in regards to details about the main exposure and outcome measures. It is unclear whether mental health was assessed in aggregate or only by types of mental health disorders. The definition of these disorders and their mechanism for confirmation of diagnosis is also lacking. While Table 1 notes that the exposure should be diagnosed clinically or by any psychological tool, this is not in the methods anywhere and should be defined in more detail. Also, how the included articles actually diagnose and define the exposure and outcome should be included in the results section. Some of this information is included in Table 2 but it is important for the reader to have this information readily available to demonstrate similarities and differences across studies. This goes for both the exposure definitions and outcome measure definition. For example, it is unclear of whether you were only interested in studies with bacteriologically confirmed cases or not. Also, can studies include individuals of any age or only adults? If so, what age range is being used to define adults.

Thank you for highlighting this. We have now expanded the Methods clarify how we are defining mental disorders and TB disease for inclusion in our review (Methods, Selection criteria, lines 56-73).

We have also clarified that no exclusions were made based on age (Methods, Selection criteria, lines 53-4). We have added detail at the start of the Results to explain how included studies measure mental disorders and TB and draw out how this varies by study type (Results, lines 121-139), and have added further details throughout the Results section as appropriate.

- Table 1 says inclusion criteria are prospective cohort studies, but Table 2 shows that retrospective cohort studies were also included.

This has now been corrected to read 'longitudinal' studies in Table 1, and in the text (Methods, Selection criteria, line 47).

- The results section notes that studies were assessed over a 50 year period, but how recent were the 10 included? Can the authors provide some more information in the text about reasons why articles

were excluded? When the authors note that subgroups of included studies make up a combined sample size (lines 112, 120, 121), it would be useful to include the percentage of the total sample size that they make up (ex. (4/10) 40% of the studies included make up 60% (361,765) of the patients included).

A sentence has been added to clarify the publication dates of the included studies (Results, lines 114-5). Reasons why articles were excluded are presented in Figure 1, with a sentence summarising key reasons for exclusion now added to the text (Results, lines 108-10). Information on the proportion of the total sample size that subgroups of included studies make up has now been added (Results, lines 126-7 and 137-9).

- Line 170—is there a 95% CI that can be reported?

This study unfortunately does not report confidence intervals.

- Quality of studies was not adequately addressed in the text of the results section.

Thank you for highlighting this, we have now added a section on study quality (Results, lines 143-9).

- It is unclear why line 184 concentrates on studies in Asia when there were also studies from other continents included?

We wanted to highlight here that although studies are included from several continents, the four cohort studies are all based in Asia. This sentence has now been rephrased to clarify this (Discussion, lines 208-9).

- Table 2 shows the critical appraisal score from the CASP and AXIS tools; there is one score that is out of 10 while all others are out of 11 and 20. Should this be out of 11?

The score out of 10 is for the one case-control study included (using the CASP checklist for case-control studies), whereas the scores out of 11 are for the cohort studies (using the CASP checklist for cohort studies) and the scores out of 20 are for the cross-sectional studies (using the AXIS tool). This is shown in full in Supplementary Table 1.

- The discussion seems to have quite a bit of repeat reporting of results which seems unnecessary. It would be good to have more discussion of the observed similarities and differences across studies, including settings/population, measuring of the exposure and outcome variables, etc. and their implication on the observed results.

Thank you for this suggestion, we have now reworked the second and third paragraphs of the discussion to discuss similarities and differences across the included studies and some potential explanations of variations in the observed results (Discussion, lines 213-230).

VERSION 2 – REVIEW

REVIEWER	Azeez, A Department of Statistics, University of Fort Hare
REVIEW RETURNED	01-Aug-2021
GENERAL COMMENTS	This manuscript meets the standard of a systematic review with the following: 1. Good literature references and sufficient field 2. Research design well-defined and identified knowledge gap 3. Method described sufficiently But impact of the study not adequately addressed.